materials science

Mo$_2$C/C composites, luffa sponge, molten salts

**Author for correspondence:**
Wanyi Zeng
e-mail: wyzeng@iue.ac.cn

This article has been edited by the Royal Society of Chemistry, including the commissioning, peer review process and editorial aspects up to the point of acceptance.

# Facile synthesis of porous Mo$_2$C/C composites by using luffa sponge-derived carbon template in molten salt media

## Minzhong Huang[1], Wanyi Zeng[2] and Ziwen Zhu[1]

[1]Advanced Materials Lab., College of Materials, Xiamen University, No. 422 Siming South Road, Xiamen, Fujian 361005, People's Republic of China
[2]Institute of Urban Environment, Chinese Academy of Sciences, No. 1799 Jimei Road, Xiamen, Fujian 361021, People's Republic of China

WZ, 0000-0001-6654-2353

Herein, we report the synthesis of a new type of porous Mo$_2$C/C composite by using luffa sponge-derived carbon template and ammonium molybdate ((NH$_4$)$_6$Mo$_7$O$_{24}$•4H$_2$O) in molten NaCl–KCl salt media. The product exhibits a higher specific surface area and three-dimensional porous structure, including macrochannels, micropores and mesopores. The desirable porous structure results from the carbon template structure and Mo$_2$C coating formed.

## 1. Introduction

Molybdenum carbide (Mo$_2$C) is a functional material that offers attractive properties, such as high hardness, a high melting point, high electrical and thermal conductivities, excellent stability and corrosion resistance. In particular, the excellent catalytic properties of Mo$_2$C, similar to noble metals [1], have increased the focus of research. Therefore, Mo$_2$C is being considered as a potential alternative to platinum (Pt) group metals in catalysis [2–4].

In order to obtain excellent performances of Mo$_2$C and extend its application fields, constructing nanostructured materials has been considered one of the best routes, which benefit from the much more active sites and shorter diffusion path [5–7]. Besides, a number of different microstructures of Mo$_2$C-based carbon composites, including nanotubes [8], nanospheres [9], hollow spheres [10] and core–shell structures [11], have been reported with enhanced performance. These reports prove that

**Figure 1.** XRD patterns of the as-prepared samples: (*a*) MCC-750, (*b*) MCC-800 and (*c*) MCC-850.

the development of a complex material to improve the performance of $Mo_2C$ is an emerging research trend.

Biological materials possess a wide variety of microstructures [12] and can be used as precursors for carbon materials with different structures. In addition, most biological materials have the advantages of low cost and environmental friendliness, so they are widely used to support various functional materials [13,14]. Among them, luffa sponge exhibits a unique porous structure with a dense and parallel arrangement [15]. It is very attractive as the precursor to prepare hierarchically porous carbon materials.

In this paper, we have successfully synthesized a new type of porous $Mo_2C/C$ composite using a luffa sponge via the molten salt method. In the process, luffa sponge acted as a carbon template and carbon source, which resulted in functional $Mo_2C/C$ composite. Thus, the product exhibited a three-dimensional porous structure, similar to the starting template, and possessed a higher specific surface area.

## 2. Materials and methods

Luffa sponge is used as the carbon template. Before carbonization, the luffa sponge was washed with water and dried at 60°C. Then the luffa sponge was placed into an argon-filled tube furnace and heated to 800°C, with a heating rate of 5°C min$^{-1}$, for 2 h. The porous carbon template was collected and used as a carbon source for the synthesis of $Mo_2C$ coating in molten salt media. Briefly, the carbon template and the ammonium molybdate (molar ratio of C/Mo = 6 : 1) were mixed with the salts (molar ratio of NaCl/KCl = 1) in an alumina crucible, and then the mixture was heat-treated under argon for 1 h. After reaction in the molten salt media, the final product was obtained by washing several times with distilled water. The products were prepared at temperatures of 750°C, 800°C and 850°C and denoted as MCC-750, MCC-800 and MCC-850, respectively.

During the preparation of $Mo_2C$ coating, ammonium molybdate was used as the Mo source and thermally decomposed into $MoO_3$. Then a chemical reaction of carbon and $MoO_3$ generated $MoO_2$ in molten salt, which further reacted with carbon to form $Mo_2C$ [16].

$$MoO_3 + C \rightarrow MoO_2 + CO_x \tag{2.1}$$

and

$$MoO_2 + C \rightarrow Mo_2C + CO_x. \tag{2.2}$$

X-ray diffraction (XRD), X-ray photoelectron spectra (XPS), scanning electron microscopy (SEM) and energy-dispersive X-ray were used for phase identification, surface composition analysis, microstructural

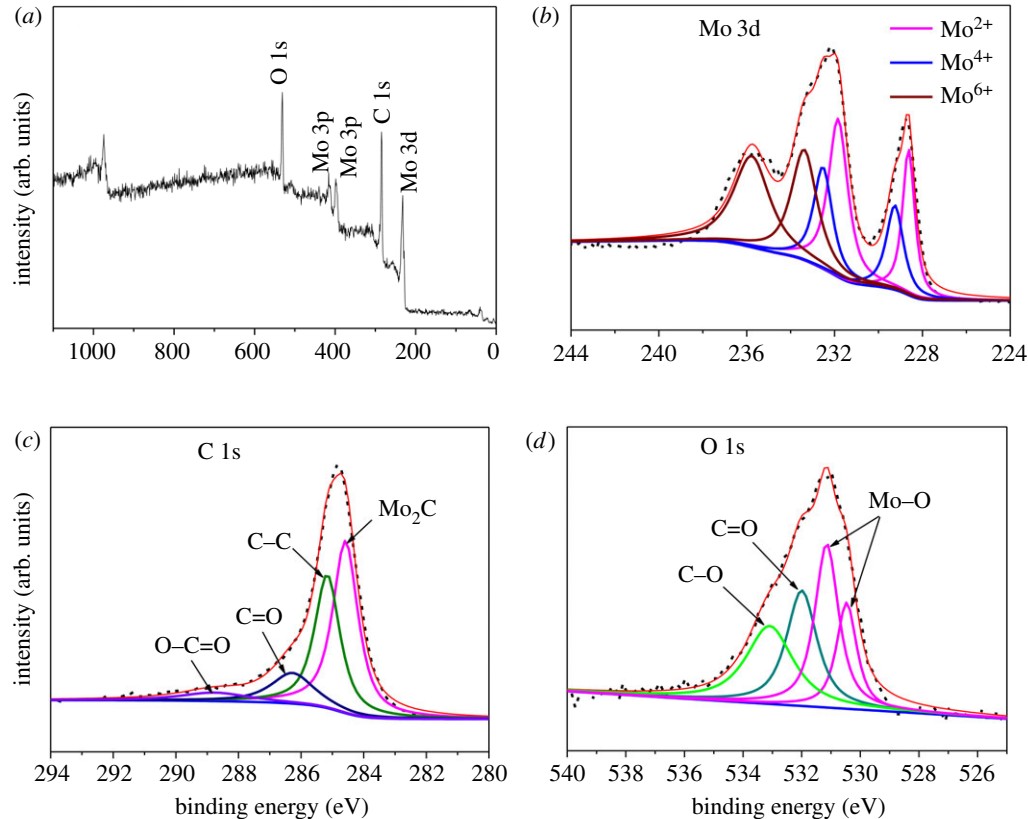

**Figure 2.** XPS spectrums of MCC-850: (*a*) survey spectrum, and high-resolution spectra for (*b*) Mo 3d, (*c*) C 1s and (*d*) O 1s.

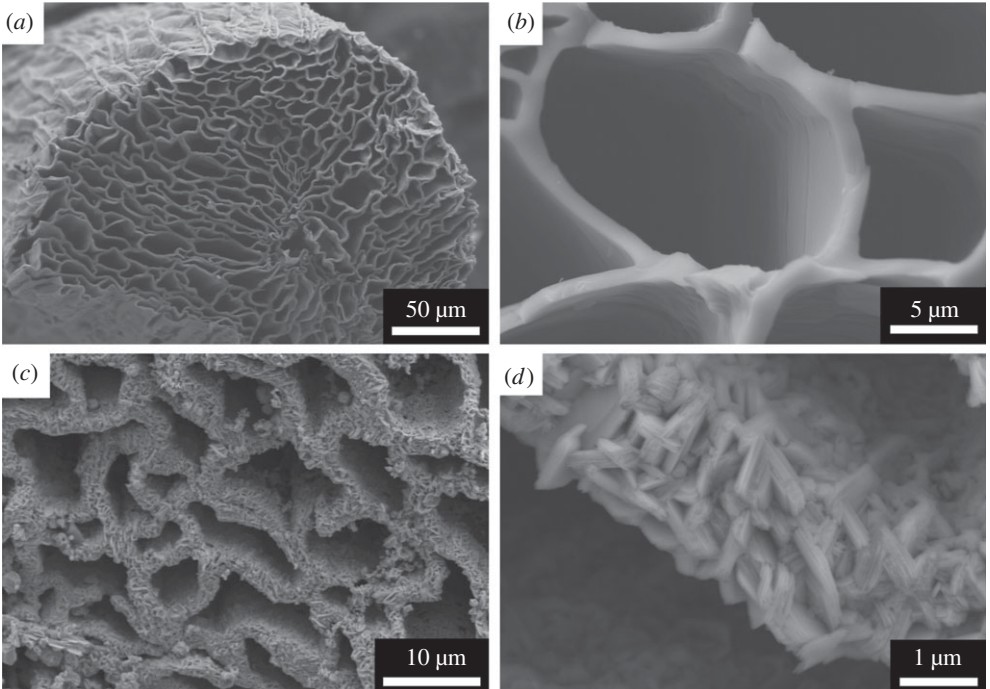

**Figure 3.** SEM images of the as-prepared samples: (*a,b*) carbon template and (*c,d*) MCC-850.

observations and component analysis, respectively. Nitrogen sorption isotherms were measured at 77 K. The specific surface area was calculated by using the Brunauer–Emmett–Teller (BET) method. The pore sizes ($D_p$) were calculated from the adsorption branches of the isotherms by using the Barrett–Joyner–Halenda model.

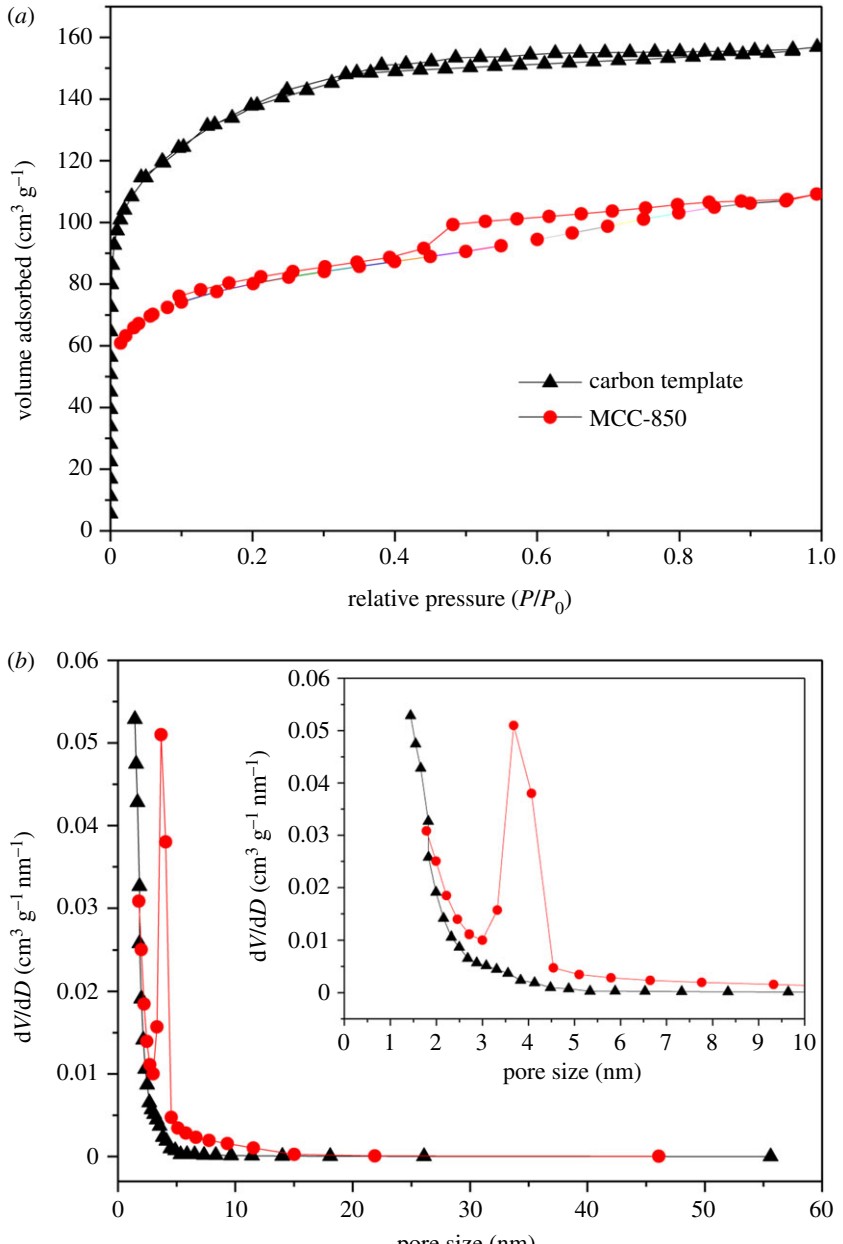

**Figure 4.** N$_2$ adsorption – desorption isotherms (*a*) and pore distribution (*b*) of carbon template and Mo$_2$C/C composites.

## 3. Results and discussion

Figure 1 presents the XRD patterns of the final product after a chemical reaction at different temperatures. After reaction at 750°C (curve a), only MoO$_2$ phases could be identified, especially the sharp characteristic peak (002) of MoO$_2$, which implies that Mo$_2$C has not been synthesized at this temperature. At 800°C (curve b), MoO$_2$ and Mo$_2$C phases coexist and a partial transformation of MoO$_2$ to Mo$_2$C has been observed. However, when the temperature was increased to 850°C (curve c), the MoO$_2$ peaks all disappeared and the Mo$_2$C peaks became more intense, which shows the lattice parameters (hexagonal P63/mmc, $a = 3.0124$ Å, $c = 4.7352$ Å) are in good agreement with that of β-Mo$_2$C in the literature [16,17]. Even though the final products are rich in carbon, we have not observed the diffraction peaks of carbon, which indicates that the luffa sponge has been carbonized into amorphous carbon during template preparation.

Taking MCC-850 as a sample, the composition and surface chemical state of the Mo$_2$C/C composite were further investigated by XPS. As shown in figure 2*a*, the survey spectrum of the sample shows distinct signals at 232.0, 284.8, 398.4, 416.0 and 531.2 eV, which can be assigned to Mo 3d, C 1s, Mo

**Table 1.** Specific surface area ($S_{BET}$), total pore volume ($V_{total}$) and average pore diameter ($D_p$) of as-prepared samples.

| sample | $S_{BET}$ ($m^2 g^{-1}$) | $V_{total}$ ($cm^3 g^{-1}$) | $D_p$ (nm) |
| --- | --- | --- | --- |
| carbon template | 495.3 | 0.201 | 1.98 |
| MCC-850 | 259.6 | 0.236 | 3.44 |

$3p_{3/2}$, Mo $3p_{1/2}$ and O 1s, respectively. Figure 2b presents a high-resolution Mo 3d XPS spectrum, which can be deconvoluted into six peaks. The peaks at 228.6 and 231.8 eV can be assigned to $Mo_2C$, and the peaks at 229.2 and 232.5 eV are related to $Mo^{4+}$, while the other two peaks at 233.3 and 235.7 eV correspond to $Mo^{6+}$ [18]. $Mo^{4+}$ and $Mo^{6+}$ can be assigned to molybdenum oxides possibly from the surface oxidation of Mo species during the process of the XPS measurement. Figure 2c shows the C 1s high-resolution XPS spectrum, in which the peak at 284.5 eV is characteristic of the Mo–C bond, whereas peaks at 285.1, 286.2 and 288.7 eV can be ascribed to C–C, C=O and O–C=O, respectively. Finally, the O 1s signal shows four peaks (figure 2d), the peaks at 530.4 and 531.1 eV belonging to the Mo–O bond, and another two (531.9 and 533.1 eV) belonging to C=O and C–O, respectively.

Figure 3 shows the cross-sectional SEM images of the as-prepared samples. After carbonization, the typical SEM image of the carbon template is shown in figure 3a, which consists of densely packed channels and exhibits a honeycomb-like porous structure. These channels are isolated by thin walls and are parallel to each other along the axial direction. The thin wall has a smooth surface and a thickness of approximately 0.5–2 μm (figure 3b). The morphological features are the result of the original texture of luffa sponge. Figure 3c,d presents the SEM images of the MCC-850 composites, prepared in molten salt media. It can be clearly observed that the surfaces of the carbon template are rough and covered with coating materials, which are $Mo_2C$ grains. Moreover, the SEM images demonstrate that the coating material has been successfully and uniformly grown on the internal wall surface; it benefits from the molybdenum oxide molecules that could be etched and dissolved in the molten salts to form a liquid solution [17]. At higher magnification (figure 3d), we can see that $Mo_2C$ grains agglomerated together with the size of submicrometre-to-micrometre. Most importantly, the original porous structure of carbon template has not been destroyed during this conversion in molten salt media.

Figure 4 presents the nitrogen adsorption–desorption isotherms and pore size distribution of the different samples. Both of the two samples have a hysteresis loop (figure 4a), but the carbon template is not obvious, which indicates that the microporous structure is dominant. Meanwhile, MCC-850 exhibits a typical type-IV absorption isotherm with a pronounced capillary condensation step, which suggests that this material has a mesoporous structure and corresponds to the pore distribution (figure 4b). The pore structure parameters of the different samples are given in table 1. It can be seen that the carbon template has a higher BET surface area of 495.3 $m^2 g^{-1}$. After $Mo_2C$ coating, the BET surface area of the MCC-850 sample decreased to 259.6 $m^2 g^{-1}$. This phenomenon can be attributed to the formation of $Mo_2C$ coating on the surface of the carbon template.

## 4. Conclusion

In conclusion, a new type of porous $Mo_2C$/C composite has been successfully synthesized using a carbonized luffa sponge as a carbon template by the molten salt method. The composites retained the shape and structure of the original template, which included packed channels and a honeycomb-like porous structure. In molten salt media, the $Mo_2C$ coating has been uniformly fabricated on the internal wall of macrochannels. The simplicity, versatility, high efficiency and low cost of the synthesis route shows promise in the fabrication of $Mo_2C$/C composites from various biological and non-biological carbon materials.

Data accessibility. Data available from the Dryad Digital Repository: https://doi.org/10.5061/dryad.2kk56sp [19].
Authors' contributions. M.Z.H. designed the study, performed the laboratory experiment and wrote the manuscript. Z.W.Z. assisted in analysing experimental data. W.Y.Z. assisted in analysing experimental data and editing the manuscript for important intellectual content, and gave the final approval for publication.
Competing interests. We declare we have no competing interests.
Funding. Financial support came from the National Natural Science Foundation of China (grant no. 51478449).
Acknowledgements. We thank the teachers of the instrument testing centre for helping to test the samples.

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
