## [Reviewer comments · Royal Society Open Science]

Review History

RSOS-190547.R0 (Original submission)

Review form: Reviewer 1

Is the manuscript scientifically sound in its present form?

Yes

Are the interpretations and conclusions justified by the results?

Yes

Is the language acceptable?

Yes

Is it clear how to access all supporting data?

Not Applicable

Do you have any ethical concerns with this paper?

No

Have you any concerns about statistical analyses in this paper?

No

Recommendation?

Accept with minor revision (please list in comments)

Comments to the Author(s)

This manuscript described the synthesis of porous Mo₂C/C composite by using luffa sponge as both structure template and carbon source in molten salts media. The biological carbon source was utilized to prepare the carbide, which is interesting. The manuscript was well established and the presentation was clear. Therefore, I would recommend its publication after the following revisions:

1. The textural character of luffa sponge seems to be important and unique for the Mo₂C/C composite, which has been described in the manuscript. However, it's not emphasized in the conclusion. It would be better if it can be included in the conclusions.
2. The authors prepared samples with molar ratio of C/Mo = 6:1, why use this ratio? And what is the reason?
3. The equation (1) and (2) should not use the equal signs.
4. The number format of reference 1 in the manuscript was not correct. In addition, there is a word "area" missing at the end of the introduction section.

Review form: Reviewer 2

Is the manuscript scientifically sound in its present form?

Yes

Are the interpretations and conclusions justified by the results?

No

Is the language acceptable?

Yes

Is it clear how to access all supporting data?

Yes

Do you have any ethical concerns with this paper?

No

Have you any concerns about statistical analyses in this paper?

No

Recommendation?

Accept with minor revision (please list in comments)

Comments to the Author(s)

In this short paper, the authors report on the synthesis of porous Mo₂C/C composites by using luffa sponge as a template. The results are interesting. However, there are some issues that the authors need to address before I can recommend its publication.

1. Can the authors elucidate a bit more about the advantages of their synthesis method of Mo₂C/C composites over the reported ones in the introduction?
2. In addition to XRD, can the authors use other method (e.g., XPS) to verify the composition of the synthesis product?

3. Can the authors comment on how the SBET can be improved?
4. It would be great if at least one practical application of the verified synthesis product can be demonstrated.
5. I cannot access the authors' data from the link they provided in the Data Accessibility section.

Decision letter (RSOS-190547.R0)

24-Apr-2019

Dear Dr Zeng:

Title: Facile synthesis of porous Mo₂C/C composites by using luffa sponge-derived carbon template in molten salt media
Manuscript ID: RSOS-190547

Thank you for submitting the above manuscript to Royal Society Open Science. On behalf of the Editors and the Royal Society of Chemistry, I am pleased to inform you that your manuscript will be accepted for publication in Royal Society Open Science subject to minor revision in accordance with the referee suggestions. Please find the reviewers' comments at the end of this email.

The reviewers and handling editors have recommended publication, but also suggest some minor revisions to your manuscript. Therefore, I invite you to respond to the comments and revise your manuscript.

Because the schedule for publication is very tight, it is a condition of publication that you submit the revised version of your manuscript before 03-May-2019. Please note that the revision deadline will expire at 00.00am on this date. If you do not think you will be able to meet this date please let me know immediately.

- 1) A text file of the manuscript (tex, txt, rtf, docx or doc), references, tables (including captions) and figure captions. Do not upload a PDF as your "Main Document".
- 2) A separate electronic file of each figure (EPS or print-quality PDF preferred (either format should be produced directly from original creation package), or original software format)
- 3) Included a 100 word media summary of your paper when requested at submission. Please ensure you have entered correct contact details (email, institution and telephone) in your user account

4) Included the raw data to support the claims made in your paper. You can either include your data as electronic supplementary material or upload to a repository and include the relevant doi within your manuscript

5) All supplementary materials accompanying an accepted article will be treated as in their final form. Note that the Royal Society will neither edit nor typeset supplementary material and it will be hosted as provided. Please ensure that the supplementary material includes the paper details where possible (authors, article title, journal name).

Best wishes,
Dr Laura Smith
Publishing Editor, Journals

RSC Associate Editor:
Comments to the Author:
(There are no comments.)

RSC Subject Editor:
Comments to the Author:
(There are no comments.)

Reviewer comments to Author:
Reviewer: 1

Comments to the Author(s)
This manuscript described the synthesis of porous Mo₂C/C composite by using luffa sponge as both structure template and carbon source in molten salts media. The biological carbon source was utilized to prepare the carbide, which is interesting. The manuscript was well established

and the presentation was clear. Therefore, I would recommend its publication after the following revisions:

1. The textural character of luffa sponge seems to be important and unique for the Mo₂C/C composite, which has been described in the manuscript. However, it's not emphasized in the conclusion. It would be better if it can be included in the conclusions.
2. The authors prepared samples with molar ratio of C/Mo = 6:1, why use this ratio? And what is the reason?
3. The equation (1) and (2) should not use the equal signs.
4. The number format of reference 1 in the manuscript was not correct. In addition, there is a word "area" missing at the end of the introduction section.

Reviewer: 2

Comments to the Author(s)

In this short paper, the authors report on the synthesis of porous Mo₂C/C composites by using luffa sponge as a template. The results are interesting. However, there are some issues that the authors need to address before I can recommend its publication.

1. Can the authors elucidate a bit more about the advantages of their synthesis method of Mo₂C/C composites over the reported ones in the introduction?
2. In addition to XRD, can the authors use other method (e.g., XPS) to verify the composition of the synthesis product?
3. Can the authors comment on how the SBET can be improved?
4. It would be great if at least one practical application of the verified synthesis product can be demonstrated.
5. I cannot access the authors' data from the link they provided in the Data Accessibility section.

Author's Response to Decision Letter for (RSOS-190547.R0)

See Appendix A.

Decision letter (RSOS-190547.R1)

22-May-2019

Dear Dr Zeng:

Title: Facile synthesis of porous Mo₂C/C composites by using luffa sponge-derived carbon template in molten salt media

Manuscript ID: RSOS-190547.R1

It is a pleasure to accept your manuscript in its current form for publication in Royal Society Open Science. The chemistry content of Royal Society Open Science is published in collaboration with the Royal Society of Chemistry.

RSC Associate Editor
Comments to the Author:
The revisions are sufficient, and the work can now be accepted.

Reviewer(s)' Comments to Author:

Appendix A

Chinese Academy of Sciences

Institute of Urban Environment

Xiamen 361021, China

Dr. Wanyi Zeng,

E-mail: wyzeng@iue.ac.cn

Tel: +86-592-6190592

Fax: +86-592-6190592

30-Apr-2019

TO: Editorial Office

Manuscript ID: RSOS-190547

Dear Editor:

Thank you for your letter dated 24-Apr-2019. This manuscript entitled “Facile synthesis of porous Mo₂C/C composites by using luffa sponge-derived carbon template in molten salt media” (RSOS-190547) has been carefully revised according to the reviewer’s comments, and the response is also attached. A point-by-point response as follows.

Responses to Reviewer 1:

1. The textural character of luffa sponge seems to be important and unique for the Mo₂C/C composite, which has been described in the manuscript. However, it's not emphasized in the conclusion. It would be better if it can be included in the conclusions.

Response: The textural character of luffa sponge has been emphasized and added to the conclusion in the revised manuscript.

2. The authors prepared samples with molar ratio of C/Mo = 6:1, why use this ratio?

And what is the reason?

Response: According to the equation (1) and (2), MoO_3 reacts with carbon to create CO and CO_2 . If just CO or just CO_2 is created, the molar ratio of C/Mo is 2:1 and 3.5:1, respectively, so we choose the molar ratio of C/Mo = 6:1 to make sure that the products is carbon surplus. In addition, the presence of carbon improves the electrical conductivity of the product.

3. The equation (1) and (2) should not use the equal signs.

Response: The equal signs have been replaced by arrow symbols in the revised manuscript.

4. The number format of reference 1 in the manuscript was not correct. In addition, there is a word “area” missing at the end of the introduction section.

Response: The incorrect number format and missing word have been verified in the revised manuscript.

Responses to Reviewer 2:

1. Can the authors elucidate a bit more about the advantages of their synthesis method of $\text{Mo}_2\text{C}/\text{C}$ composites over the reported ones in the introduction?

Response: Compare with the reported in the introduction, the advantages of synthesis method of $\text{Mo}_2\text{C}/\text{C}$ composites in this work as follows:

(1) We use carbonized luffa sponge as a carbon template, it is low cost and environmental friendliness.

(2) The Mo_2C grain can uniformly fabricated on the carbon template by molten salt method, and the $\text{Mo}_2\text{C}/\text{C}$ composites can retained the shape and structure of the original template.

(3) The preparing method is simple, highly efficient and can extend to other carbon materials for $\text{Mo}_2\text{C}/\text{C}$ composites.

2. In addition to XRD, can the authors use other method (e.g., XPS) to verify the composition of the synthesis product?

Response: The XPS spectrums of the product has been added to the revised manuscript (Fig. 2).

3. Can the authors comment on how the SBET can be improved?

Response: In this composite, Mo₂C coating is significant factors to influence the SBET, by controlling the Mo₂C grain morphology and grain size in different process parameters, it may be possible to improve the SBET of the product.

4. It would be great if at least one practical application of the verified synthesis product can be demonstrated.

Response: In this paper, we mainly discussed the preparation method and characterization of the product, about exploring the application performance of the product will be the focus of our next work.

5. I cannot access the authors' data from the link they provided in the Data Accessibility section.

Response: The link in the Data Accessibility section is available, we have confirmed it.

Finally, thanks again to the editors and reviewers for giving me the opportunity to publish my paper in Royal Society Open Science.